# Calcium Regulates HCC Proliferation as well as EGFR Recycling/Degradation and Could Be a New Therapeutic Target in HCC

**DOI:** 10.3390/cancers11101588

**Published:** 2019-10-18

**Authors:** Teresa Maria Elisa Modica, Francesco Dituri, Serena Mancarella, Claudio Pisano, Isabel Fabregat, Gianluigi Giannelli

**Affiliations:** 1Department of Biomedical Science and Human Oncology, Università degli Studi di Bari Aldo Moro, 70121 Bari, Italy; modica.elisa@libero.it; 2Biogem S.C.A.R.L., 83031 Ariano Irpino (AV), Italy; 3IRCCS Saverio De Bellis Castellana Grotte, 70013 Bari, Italy; francesco.dituri@irccsdebellis.it (F.D.); serena.mancarella@irccsdebellis.it (S.M.); 4Biogem S.C.A.R.L., 83031 Ariano Irpino (AV), Italy; claudio.pisano@biogem.it; 5Bellvitge Biomedical Research Institute (IDIBELL) L’Hospitalet, 08907 Barcelona, Spain; ifabregat@idibell.cat; 6Faculty of Medicine and Health Sciences, University of Barcelona, 08907 Barcelona, Spain; 7Oncology Program, CIBEREHD, Instituto de Salud Carlos III, 28029 Madrid, Spain

**Keywords:** HCC, EGFR degradation, AZD9291, calcium ions, BAPTA_AM

## Abstract

Calcium is the most abundant element in the human body. Its role is essential in physiological and biochemical processes such as signal transduction from outside to inside the cell between the cells of an organ, as well as the release of neurotransmitters from neurons, muscle contraction, fertilization, bone building, and blood clotting. As a result, intra- and extracellular calcium levels are tightly regulated by the body. The liver is the most specialized organ of the body, as its functions, carried out by hepatocytes, are strongly governed by calcium ions. In this work, we analyze the role of calcium in human hepatoma (HCC) cell lines harboring a wild type form of the Epidermal Growth Factor Receptor (EGFR), particularly its role in proliferation and in EGFR downmodulation. Our results highlight that calcium is involved in the proliferative capability of HCC cells, as its subtraction is responsible for EGFR degradation by proteasome machinery and, as a consequence, for EGFR intracellular signaling downregulation. However, calcium-regulated EGFR signaling is cell line-dependent. In cells responding weakly to the epidermal growth factor (EGF), calcium seems to have an opposite effect on EGFR internalization/degradation mechanisms. These results suggest that besides EGFR, calcium could be a new therapeutic target in HCC.

## 1. Introduction

Hepatocellular carcinoma (HCC) is the most common primary liver malignancy in adults and the third most common cause of cancer death worldwide [1,2]. It is rarely detected early and is often fatal within a few months of diagnosis. Among the risk factors for the disease are hepatitis C and B virus infection, alcoholic liver disease, liver cirrhosis, tobacco smoking, and obesity [3]. Until now, surgery has played a central role in the treatment of liver cancer. However, HCC recurrence after surgical treatment is frequent and the long-term prognosis of patients with HCC is generally poor.

The disease has a complex molecular pathogenesis in which several signaling pathways could be involved [4,5,6]. In particular, various studies have demonstrated that growth factors, acting in part through intracellular calcium ([Ca^2+^]i) release [7] play a pivotal role, and their signaling pathways are known targets for therapeutic approaches [4,5,6]. Although its multistep development, multifactoriality, heterogeneity, and the interactions between integrins and extracellular matrix proteins make the correct tumor subclasses difficult to classify and stage, changes at molecular level can often be observed in HCC versus non-tumoral liver tissue.

HCC cells encode for several RTKs, including the epidermal growth factor (EGF) receptor (EGFR). Its functions involve glucose and lipid metabolism [8,9], liver fibrosis [10], regeneration [11], as well as hepatocellular carcinogenesis [11,12,13]. EGFR was first discovered in 1959 as a cell proliferation-promoting factor and was soon ascribed a proto-oncogene function. Besides activating EGFR mutations (EGFRm^+^) that are sufficient to induce multi-drug resistance (MDR), one of the most frequently observed alterations is the overexpression of wild type EGFR, as well as EGF oversecretion [14,15,16]. Some mutations also allow the receptor to escape downregulation by endocytosis [17].

Due to its key role in oncogenesis, EGFR-targeted therapies have been developed. The first class of therapy includes humanized monoclonal antibodies against the EGFR extracellular domain designed to block ligand binding [18]. The second class includes tyrosine kinase inhibitors (TKIs) that are ATP mimetics that can bind to the receptor’s kinase pocket preventing signal transduction [19]. Approved TKIs include Erlotinib, Gefitinib, and Lapatinib. Overexpression of EGFR has been observed in around 40% to 70% of HCC [20,21,22,23]. The efficacy of EGFR antagonists against HCC has been demonstrated in cancer cell lines and animal models. Gefitinib can inhibit growth and intrahepatic metastasis of implanted murine HCC [24,25,26]. Gefitinib has been evaluated in the treatment of patients with lung cancer and other tumors [26,27,28,29] and is able to reduce the onset of HCC tumors on cirrhosis via inhibiting the EGFR pathway [26]. Moreover, it inhibits cell growth in all human liver cancer cell lines [30], inducing apoptosis and cell cycle arrest [24].

However, due to the fact that after a good initial response to first-generation TKIs, non–small cell lung cancer (NSCLC) patients with EGFR-activating mutations (in particular T790M) ultimately undergo disease progression, it has been necessary to develop a novel, irreversible EGFR TKI that features selectivity against sensitizing and T790M-resistant mutant forms versus wild-type forms of EGFR. AZD9291 is a mono-anilino-pyrimidine compound that covalently binds with Cys797 in EGFR, inhibiting cell proliferation in vitro and inducing tumor regression in vivo in xenograft models [31,32,33].

For EGFR activation to occur, ligand binding, EGFR transphosphorylation of various tyrosine residues on the intracellular C-terminal tail, and its dimerization are crucial events. The large number of signaling pathways include the ERK MAPK, PI3K-AKT, SRC, PLC-1-PKC, JNK, and JAK-STAT pathways. InsP3-mediated Ca^2+^ mobilization is triggered by the binding of growth factors to receptor tyrosine kinases (RTKs). Once growth factors bind to their specific receptor, membrane-associated PLC is activated. In particular, immediately after EGF binding to EGFR, PLCγ become activated and thus able to hydrolyze the plasma membrane as well as nuclear PIP2, to generate InsP3 and Ca^2+^ release both into the cytoplasm and into the nucleoplasm [34,35].

Calcium waves spread between hepatocytes through gap junctions [36,37] to allow intralobular cell-cell synchronization. In hepatocytes, calcium oscillations are induced by EGFR activation, and in turn, calcium signals control distinct physiological hepatic processes such as bile secretion [38], glucose metabolism [39,40,41], cell proliferation [42,43], progression of the cell cycle [44,45,46,47,48], liver regeneration [49,50,51], and apoptosis [52,53,54]. Moreover, the importance of calcium as an EGF mediator in hepatic cells is underlined by the fact that Ca^2+^ signaling is regulated differently in the nucleus and cytosol, providing a mechanism for the independent regulation of Ca^2+^-dependent processes in these cellular compartments [55].

In accordance with the multiple hepatic functions of calcium, dysregulation of calcium signaling is a hallmark of both acute and chronic liver diseases. In particular, increased cytosolic amounts of calcium can have an impact on the anabolic/catabolic balance of the cell, while perturbations of mitochondrial calcium account for cell life or death through apoptosis [56]. The liver is a highly specialized organ whose cells maintain constant communication to achieve the correct performance of the whole organ function. Thus, spatial and temporal calcium waves should be considered at subcellular, cellular, and tissue level. In this context, calcium modulation is an emerging strategy in the treatment of various acute and chronic liver insults.

The goal of this work has been to devise an alternative hepatocellular carcinoma treatment that can sustain the approved TKIs therapy and at the same time reduce TKIs dosage and hence side effects. A new strategy for hepatocarcinoma treatment could be to split these two processes, EGFR activation and calcium wave, that are so tightly coupled.

## 2. Results

### 2.1. EGFR Sequencing

Sequencing of EGFR in HUH-7, HUH-6, Hep3B, and HepG2 cell lines demonstrated that EGFR was expressed in the wild form in all the HCC cell lines tested.

### 2.2. EGFR Signaling and Its Inhibition

EGFR expression and phosphorylation were tested by western blot following stimulation with EGF (100 ng/mL) for 30 min in the presence and in the absence of FBS. EGF induced pERK and pAKT upregulation in all the cell lines analyzed, as expected. However, HUH-6 cells seemed to be less sensitive to EGF than the other cell lines analyzed (Figure 1; Appendix A).

To better understand the IC50 effect of Gefitinib (GEF) and AZD9291 (AZ) EGFR inhibitors (listed in Table 1) in signaling, starved cells were treated for 3 h with GEF IC50 or AZ IC50 and DMSO as control. GEF or AZ treatment switched off EGFR, ERK, and AKT phosphorylations in all cell lines analyzed. EGF was not able to rescue AKT and ERK phosphorylation following GEF or AZ EGFR inhibition (Figure 2; Appendix A).

Moreover, as observed in Figure 3, both drugs (GEF and AZ) blocked the activated EGFR signaling, after as little as 30 min of incubation, in all the cell lines analyzed, even though the EGFR phosphorylated form was still present. Unlike in the other cells, in HUH-6 cells both GEF and AZ had already reduced the EGFR phosphorylation within 30 min, but the downstream pathways were only weakly affected as compared to the other cell lines.

As described in literature, 3 or 6 h later, EGF-stimulated cells (DMSO lane in Figure 3) undergo EGFR internalization and lysosomal degradation (a phenomenon called “EGF-dependent EGFR degradation”, as indicated by the total EGFR (EGFR TOT) level reduction) [57,58,59,60,61,62,63]. The same was not observed in the HUH-6 cell line that showed an even more robust EGFR phosphorylation until 6 h, followed by no reduction of EGFR levels.

At 3 and 6 h, EGF-stimulated cells treated with EGFR inhibitors showed a reduced internalization/degradation of the receptor compared to untreated cells, with a consequent stabilization of the receptor in its inactive form [60,64,65], once again except for the HUH-6 cell line. It may be speculated that the internalization/degradation mechanism in the HUH-6 cell line is different from that observed in the other cell lines (Figure 3; Appendix A). 

In HepG2 and HUH-6 cell lines EGFR trafficking seems to be less evident than in HUH-7 and Hep3B cells. Moreover, it is noteworthy that treatment with GEF in Hep3B cells leads to a rescue of pAKT within 3 to 6 h, as seen in Figure 3 (see also Figure 2). 

As expected, following EGFR signaling switch-off by GEF or AZ, the proliferation of all the cell lines analyzed was negatively affected by both drugs (Figure 4).

In conclusion, both AZ and GEF act on the same pathways downstream of EGFR (pAKT and pERK) and both of them are able to switch off already activated EGFR pathways, as may be expected in the in vivo context.

However, AZ activity was stronger than GEF activity even at lower concentrations, both in signaling and in proliferation assays, as indicated also by their IC50 values in Table 1.

### 2.3. EGFR Signaling and Calcium Chelators

Preliminary proliferation assays carried out on these HCC cell lines in the presence of EGF added to the serum-free medium showed an unexpected regulation of cell growth. Results suggested two possibilities: besides EGF, the component responsible could be produced by cells, and released into the culture medium (with a paracrine and antiproliferative activity), or might be already present in the culture medium and consumed over time (with a pro-proliferative activity). One of these last factors was calcium. Therefore, it was hypothesized that calcium ions could be actively involved in regulating EGFR-dependent HCC cells growth.

In order to investigate this hypothesis, HUH-7 and HUH-6 cell line proliferation was measured in starved cells (0% FBS), treated or not with EGF, in the presence of 2 mM EDTA solution as calcium chelator (Figure 5). 24 h of EDTA treatment negatively affected cell number in both cell lines within 72 h. On the contrary, 0.5% DMSO tended to increase the number of cells, mostly when added with EGF.

As widely acknowledged in literature, DMSO can induce transient water pores in cell membranes, increasing permeability, thus Ca^2+^ can easily flow through these pores from the medium to the cytosol [66,67,68,69].

The EDTA effect was observed also at molecular level by western blot on HUH-7 cells treated or not with 2 mM EDTA for 6 and 24 h (Figure 6; Appendix A). Proliferative inhibition was confirmed also by a Cyclin D1 reduction, especially within 24 h of EDTA treatment. Following calcium subtraction EGF addition did not rescue pERK nor Cyclin D1 levels as early as 6 h, even though the pEGFR level was still high, suggesting that calcium is necessary for EGFR signaling propagation. Notably, within 6 h EDTA was able to induce a sustained EGFR downmodulation as compared to EGF alone. After 24 h, EGF-dependent EGFR degradation was almost complete even without EDTA.

The effect of EDTA on pAKT 24 h later was impressive. AKT phosphorylation dramatically increased, probably to counteract the EDTA-triggered apoptotic stimulus (Figure 6A). DMSO was also used as positive control. As expected, 24 h of 0.5% DMSO treatment upregulated pERK and increased the Cyclin D1 levels more than EGF alone, indicating that intracellular free Ca^2+^ acts through the ERK pathway (Figure 6B).

These results indicated that calcium ions are involved in the proliferative capability of HCC cell lines, as well as in EGFR degradation (calcium subtraction induced EGFR degradation within 6 h in an activated system).

To rule out the possible involvement of apoptotic signals triggered by EDTA, we replaced EDTA with the less toxic EGTA and examined AKT phosphorylation (pAKT) levels at a later time (24 h). Proteins extracted from cells treated with EDTA were loaded as positive control. Molecular analysis on pAKT levels excluded any apoptotic effect after 24 h of EGTA treatment (Figure 7C; Appendix A). Moreover, also in this case the results obtained confirmed the calcium involvement. HUH-7 cells fate resulted dependent on calcium depending on their starting proliferative status. More in detail, in actively proliferating cells (10% FBS (48 h)) EGTA treatment reduced proliferation (Figure 7A), while CaCl_2_ addition promoted cell proliferation and therefore cell cycle progression. On the contrary, in non-proliferating cells (serum-free (SF) medium (48 h)), calcium addition halved the number of viable cells, and as a consequence, EGTA in the presence of calcium rescued the number of viable cells (Figure 7B).

On the basis of these results, we tested EGTA treatment in association with the two EGFR inhibitors, GEF and AZ, at their respective IC50 values. In proliferating cells (whose proliferative capability is shown by the increased number of viable cells in the CTR medium bar versus the *T0* bar), the combined treatment reduced the proliferative capability of cells using either GEF and AZ (Figure 8A). Conversely, as expected, in non-proliferating cells (the CTR medium bar and *T0* bar displayed no differences) EGTA combined with either drug led cells to maintain or even slightly increase their proliferative capacity, reverting the trend observed in proliferating cells (Figure 8B). 

However, both calcium chelators, EDTA and EGTA, do not only act on calcium ions outside the cell, but are also able to form complexes with different cations. For these reasons, we focused on the cell-permeable and specific intracellular Ca^2+^ chelator BAPTA-AM in order to manipulate the intracellular Ca^2+^ free levels.

In the HUH-7 cell line, 6 h of 10 µM BAPTA_AM treatment reduced the EGF-induced Cyclin D1 increase, even though the EGFR phosphorylation was still sufficient.

Moreover, in BAPTA_AM treated cells, after 6 h pAKT was higher than in EDTA treated cells, but within 24 h, in BAPTA_AM treated cells, it returned to the basal level and was strongly increased in EDTA samples, indicating that BAPTA_AM does not trigger the apoptotic pathway (Figure 9A,B; Appendix A).

It is noteworthy that within 6 h, the combined treatment, BAPTA_AM and EGF, was able to induce a greater EGFR level reduction than EGF stimulation alone, emphasizing the important role of calcium ions in EGFR recycling. Thus, calcium subtraction triggered EGFR lysosomal degradation (Figure 9A,B; Appendix A).

On the contrary, HUH-6 cells treated with BAPTA_AM and EGF behave differently from what was observed in HUH-7 cells. In line with what is shown in Figure 3, BAPTA_AM and EGF treatment increased the levels of EGFR within 6 h, suggesting that calcium is necessary for degradation (instead of recycling) of the EGFR activated form (Figure 9C; Appendix A).

In order to confirm the EGFR degradation through proteasome machinery, the proteasome machinery inhibitor MG132 was used. The HUH-7 and HepG2 cell lines reduced the EGFR levels 6 h of EGF stimulation, as seen above. BAPTA_AM further emphasized this result whereas MG132 rescued the EGFR levels. These results indicate that in both cell lines the EGF-induced EGFR degradation is reinforced by calcium subtraction after as little as 6 h, and it is triggered by proteasome.

Once again, HUH-6 cells showed the opposite effect. In EGF-stimulated cells, BAPTA_AM increased the EGFR levels, as if the reduction of calcium level blocked degradation in favor of recycling and MG132 inverted the effect, reducing the levels of EGFR (Figure 10; Appendix A).

In terms of proliferation, 48 h treatment with 10 µM BAPTA_AM on HUH-7 as well as HUH-6 cells reduced cell viability. Moreover, if used with half doses of GEF or AZ (IC50/2) in a combined administration, BAPTA_AM further reduced viable cells. The same result was observed in Hep3B and HepG2 cell lines (Figure 11A–D).

## 3. Discussion

The main aim of this study was to find a second target involved in the EGFR pathway to be targeted alone or in combination with current therapies (TKIs) in order to reduce TKIs dosage and hence their side effects, as well as the multi-drug resistance often acquired by cancer cells. 

We first tested the AZD9291 on the HCC cell lines studied, showing that its activity was stronger than that of Gefitinib, even at lower doses. AZD9291 has been developed as a third-generation irreversible inhibitor with selectivity against T790M mutant versus wild type EGFR. However, on the HCC cell lines analyzed, the inhibition of proliferation was greater with AZD9291 than Gefitinib, even though all the cell lines tested harbored the wild type form of EGFR. As already demonstrated for Gefitinib, AZD9291 also acts on AKT and ERK pathways.

On the basis of our previous observations, we noticed that EGF-dependent HCC proliferation was governed also by a second factor. After having excluded the presence of some apoptosis-induced factors produced and released by cells, we postulated that calcium ions could be involved in this process. To investigate this possibility, we used different calcium chelators. First results observed using EDTA showed a great involvement of calcium both in EGFR-dependent proliferation and in EGFR signaling. To corroborate these results we treated cells with DMSO. As already described in literature, DMSO can induce water pores in dipalmitoyl-phosphatidylcholine bilayers through which ions can penetrate inside the cell [70]. Through these pores, it becomes easier for calcium ions to pass in and out of the cell and thus to function as positive regulators of cell proliferation [71,72]. In line with this, DMSO treatment enhanced proliferation, activating the pro-proliferative pathways, especially when added together with EGF.

However, EDTA upregulated the AKT phosphorylation independently of EGF, demonstrating its high apoptotic potential [73,74]. For this reason, we moved on to the EGTA calcium chelator. Unlike EDTA, EGTA showed no toxicity. It was noteworthy that within only 6 h of combined EGTA plus EGF treatment, the levels of EGFR were markedly reduced, more than by EGF alone. After 24 h of EGF treatment alone the EGFR downmodulation was almost complete. This stimulus-induced trafficking is already known as EGF-dependent EGFR internalization and is aimed at regulating the timing of EGFR signaling. In this work, by extracellular calcium withdrawal, we were able to anticipate the phenomenon from 24 to 6 h. Our results were confirmed using the more specific cytoplasmic calcium ions chelator BAPTA_AM. 

Taken together, these results show that intracellular free calcium, necessary for EGFR signal propagation to occur inside the cell, is also a key regulator of cell cycle progression or apoptosis in HCC cell lines, depending on their proliferative status at that specific time point. More in detail, calcium enhances proliferation in already proliferating HCC cells whereas it pushes non-proliferating (G0) HCC cells towards apoptosis.

Besides its role in the cell cycle, we observed for the first time that after as little as 6 h treatment, calcium is essential also for the active EGFR fate: recycling or degradation. To distinguish between the two EGFR events, proteasome inhibition by MG132 was necessary. More in detail, in EGF-sensitive cells, such as HUH-7, HepG2, and Hep3B, intracellular calcium subtraction facilitates EGF-induced EGFR degradation, that is otherwise visible only much later (24 h), indicating that in these cells recycling is calcium-dependent. Conversely, HUH-6 cells did not only seem to be less sensitive to EGF stimulus but also showed an increased EGFR-recycling following BAPTA_AM treatment, suggesting that in this cell line calcium positively regulates EGF proteasomal degradation rather than recycling.

However, in both cases combined therapy with Gefitinib or AZD9291 and BAPTA_AM reduced the HCC cell viability, also with half doses of TKIs. For the purposes of avoiding MDR acquisition by HCC, this result is really significant if we take into consideration the fact that, as demonstrated, a sustained increase of intracellular calcium is actively related to MDR in HCC and the inhibition of calcium enhanced the efficacy of chemotherapy [75]. Moreover, early EGFR internalization and its consequent signaling down-modulation could be of great relevance in HCC treatment. Indeed, as shown in Figure 8A,B), in G0 phase cells calcium subtraction tended to reduce the drug activity. On the contrary, in actively proliferating cells, TKIs increased their antiproliferative ability following calcium chelation. Considering that the in vivo tumoral condition is closer to an actively proliferating than to a quiescent system, it may be that TKIs could be more effective if administered together with an intracellular calcium chelator. However, calcium chelator administration in vivo in xenograft HCC models requires further accurate, close investigation in order to guarantee a correct delivery of the drug and avoid its release in body districts strongly dependent on calcium for their function (such as the muscle or cardiac systems).

## 4. Materials and Methods

### 4.1. Cell Culture

Human hepatocellular carcinoma cell lines HUH-6 and HUH-7 were cultured in DMEM medium (L0104-500 Microgem laboratory research, 80131 Naples, Italy) whereas Hep3B and HepG2 were cultured in MEM medium (L0415-500 Microgem laboratory research) at 37 °C in a humidified atmosphere of 5% CO. Both medium were supplemented with 1% penicillin-streptomycin (L0022-500 Microgem laboratory research) and 10% fetal bovine serum (L1810-500 Microgem laboratory research).

### 4.2. Protein Extraction 

Cellular protein was extracted using RIPA lysis and extraction buffer (Sigma-Aldrich, St. Louis, MO, USA) containing a protease and phosphatase inhibitors. Cell pellets were lysed for 30 min, and samples were centrifuged at 12,000 rpm for 20 min at 4 °C in a microcentrifuge. The supernatant liquid was collected in new Eppendorf tubes and stored at −20 °C.

Protein concentration was measured using the Bradford protein assay (Bio-Rad Laboratories Inc., Hercules, CA, USA).

### 4.3. Cell Viability Assay

Cell viability was measured using the Sulforhodamine B (SRB) assay, which is based on the stoichiometric binding of SRB dye to proteins under mild acidic conditions and its subsequent extraction under basic conditions. The amount of dye extracted is a proxy for cell mass and thus the number of cells in a sample. The absorbance of the dye in solution is measured at OD 565 nm using an automated microplate reader (Perkin Elmer, Waltham, MA, USA). Sulforhodamine B sodium salt (S9012) was purchased from Sigma.

### 4.4. Antibodies and Drug Formulations

The following antibodies were used according to the protocols supplied by the manufacturers: anti-pERK1/2 (#9101, Cell Signaling Technology, Danvers, MA, USA), anti-ERK1/2 (#4695, Cell Signaling Technology), anti-GAPDH (ENM0040, Elabscience Biotechnology Inc., Houston, TX, USA), anti-pEGFR (12A3) (sc-57542, Santa Cruz Biotechnology Inc., Dallas, TX, USA), anti-EGFR (C-2) (sc-377229, Santa Cruz Biotechnology Inc.), anti-pAKT (#9271, Cell Signaling Technology), anti-AKT (#9272, Cell Signaling Technology), anti-Cyclin D1 (sc-753, Santa Cruz Biotechnology Inc.), anti-p21 (sc-397, Santa Cruz Biotechnology Inc.), and anti-beta Tubulin (Sigma), anti-alpha actin (Sigma). All the secondary antibodies (HRP-conjugated anti-rabbit and anti-mouse) were purchased from Santa Cruz Biotechnology Inc.

Drug formulations: gefitinib (ZD1839) and MG132 (S2619) were purchased from Selleckchem; AZD9291 was obtained from AstraZeneca; BAPTA_AM (HB0981) was from HelloBio.

Recombinant Human EGF (AF-100-15) was from PeproTech (London, UK).

### 4.5. Western Blot Analysis

40 μg proteins per sample were separated using 4–20% SDS-PAGE and transferred to 0.2 µm nitrocellulose Trans-blot turbo^TM^ membranes (Bio-Rad Laboratories Inc.) using the Bio-Rad electrotransfer system (Bio-Rad Laboratories Inc.). The membranes were blocked with blocking buffer (mixed 5% non-fat dry milk, 150 mM NaCl, 0.1% Tween-20 and 20 mM Tris-HCl and adjusted to a pH of 7.6) for 1 h at room temperature and probed with specific primary antibodies at 4 °C overnight. The protein bands were detected with HRP-conjugated secondary antibodies for 1 h at room temperature using custom-made ECL^TM^ Prime Western Blotting Detection Reagents (Amersham GE, Little Chalfont, UK). The Image Lab^TM^ software digital imaging system ChemiDoc^TM^ XRS+ (Bio-Rad Laboratories Inc.) was used to detect the target protein on immunoblot nitrocellulose membranes.

### 4.6. Statistical Analysis

Plotted values are shown as means ± standard deviation. Statistical significance of the results was determined using the two-tailed unpaired Student’s *t* test to determine whether the two datasets were significantly different. A value of *p* < 0.05 was considered significant.

For cell survival data of Figure 4, data from GEF and AZ treatments were compared, fitting a linear model for each subgroup of samples. The differences between estimated coefficients were assessed through the generation of a model which also contemplates an interaction term between the variable days and drug. The p-value obtained provides confidence about the generalizability of the differences in linear trends observed for GEF and AZ treatments.

## 5. Conclusions

Considering the plethora of calcium activities in the liver, as well as the fact that calcium channels are overexpressed in many HCC, where calcium plays a role in inducing MEK/ERK-triggered proliferation, calcium manipulation in HCC cells may have a therapeutic potential in preventing tumor growth.

Moreover, combining BAPTA_AM treatment with EGFR inhibitors could help, on one hand to reduce drug doses and thus elevated toxicity, and on the other hand, to overcome EGFR acquired resistance to EGFR-TKIs.

This kind of treatment fits into the frame of more accurate therapy, along the lines of constantly developing personalized medicine. 

## Figures and Tables

**Figure 1 cancers-11-01588-f001:**
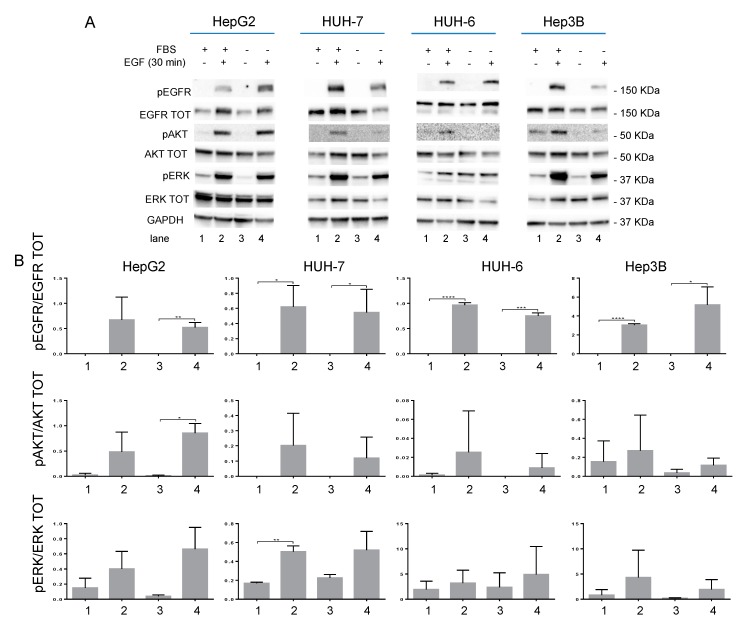
(**A**) Western blot analysis of EGFR pathway activation in HepG2, HUH-7, HUH-6, and Hep3B cell lines. (**B**) Densitometric analysis calculated by image lab software of the western blot shown in Figure 1A; numbers in the abscissa refer to the corresponding lane in panel A. *p* value < 0.05 (*); *p* value < 0.01 (**); *p* value < 0.001 (***); *p* value < 0.0001 (****).

**Figure 2 cancers-11-01588-f002:**
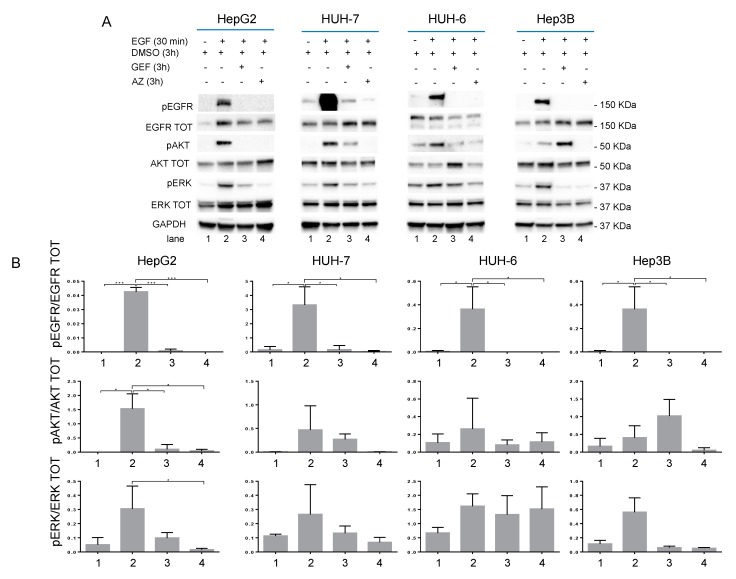
(**A**) Western blot analysis of HepG2, HUH-7, HUH-6, and Hep3B starved cell lines treated with GEF IC50 or AZ IC50 (as indicated in Table 1) (DMSO as control) for 3 h before stimulation with 100 ng/mL of EGF for 30 min. Panel (**B**) shows the densitometric analysis calculated by image lab software of the western blot shown in Figure 1A; numbers in the abscissa refer to the corresponding lane in panel A. *p* value < 0.05 (*); *p* value < 0.01 (**); *p* value < 0.001 (***).

**Figure 3 cancers-11-01588-f003:**
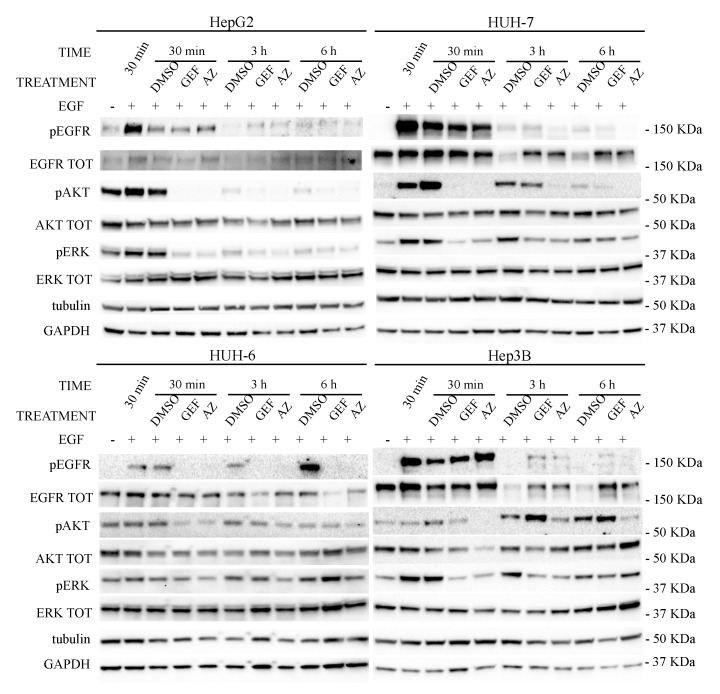
Western blot panels of HepG2, HUH-7, HUH-6, and Hep3B starved cell lines stimulated with 100 ng/mL of EGF for 30 min before and during treatment with GEF or AZ IC50 (as indicated in Table 1) (DMSO as control). Treatments were performed for 30 min, 3 h and 6 h.

**Figure 4 cancers-11-01588-f004:**
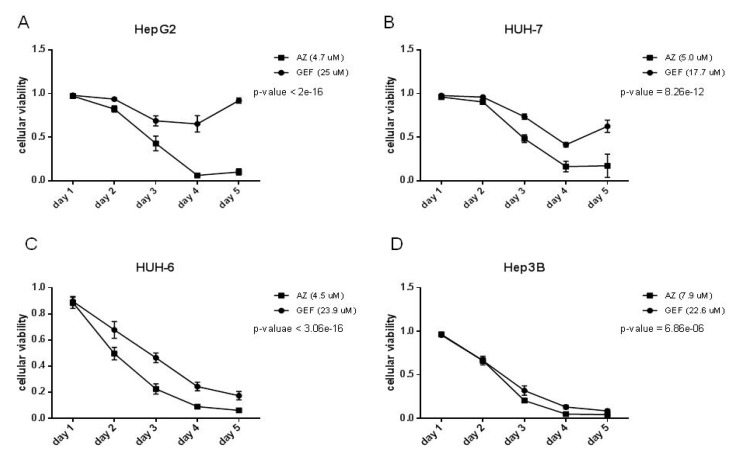
Five days proliferation assay of (**A**) HepG2, (**B**) HUH-7, (**C**) HUH-6, and (**D**) Hep3B cell lines. 10% FBS culture medium was added with GEF IC50 or AZ IC50 (as indicated in Table 1), DMSO as control. Data are plotted in the graph as normalized by DMSO.

**Figure 5 cancers-11-01588-f005:**
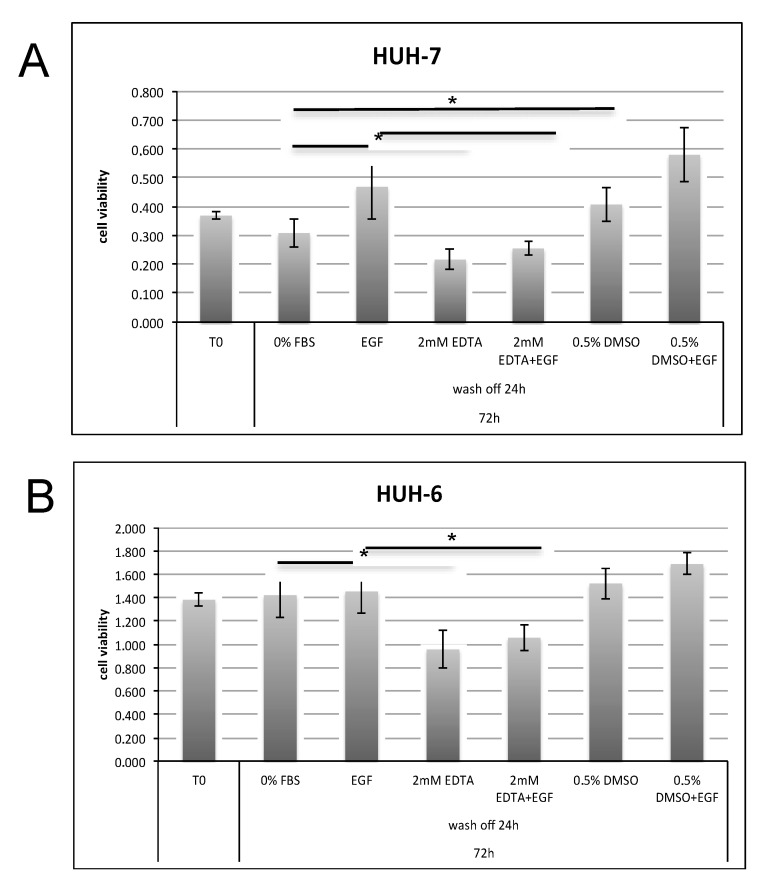
Starved HUH-7 (**A**) and HUH-6 (**B**) cell lines (T0) were left untreated (0% FBS as CTR) or treated with 100 ng/mL EGF, 2 mM EDTA, 0.5% DMSO or combined compounds (as indicated in the graphs) for 24 h. After 24 h of treatment, medium was replaced and cells were maintained in culture for a further 48 h in serum-free medium (0% FBS) with or without EGF, on the basis of the previous treatment. The proliferative capability of the cells was evaluated after a total time of 72 h. *p* value < 0.05 (*).

**Figure 6 cancers-11-01588-f006:**
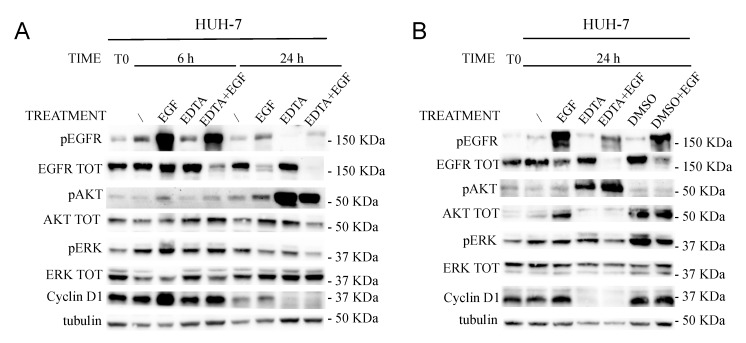
Starved HUH-7 cells (T0) were left untreated (/) (0% FBS as CTR) or treated with 100 ng/mL EGF, 2 mM EDTA, 0.5% DMSO, or combined compounds (as indicated in the figures). The cell signaling cascade was analyzed by western blot after 6 h (**A**,**B**) and 24 h (**B**).

**Figure 7 cancers-11-01588-f007:**
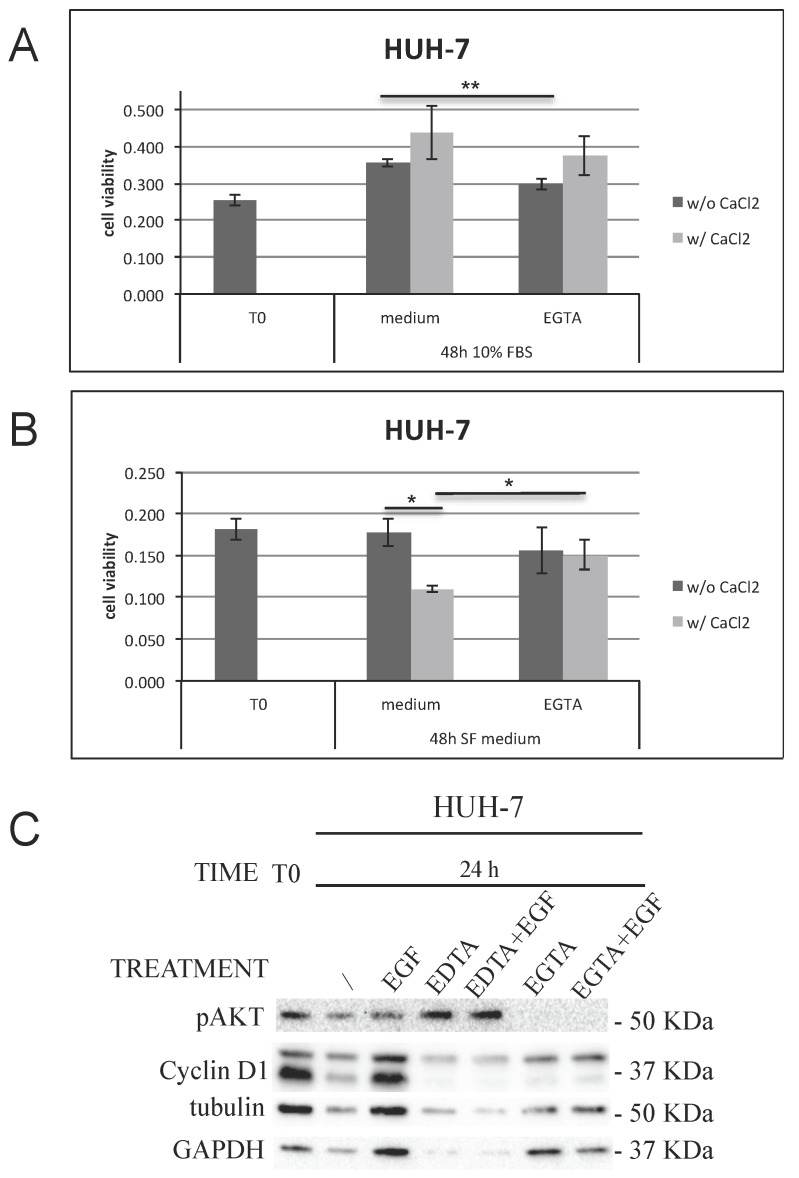
HUH-7 not starved (10% FBS (**A**)) or starved (SF medium (**B**)) cells were left untreated (medium as CTR) or treated with 2 mM EGTA for 48 h in the presence or absence of CaCl2. The cell signaling cascade was analyzed by western blot after 24 h (**C**). *p* value < 0.05 (*); *p* value < 0.01 (**).

**Figure 8 cancers-11-01588-f008:**
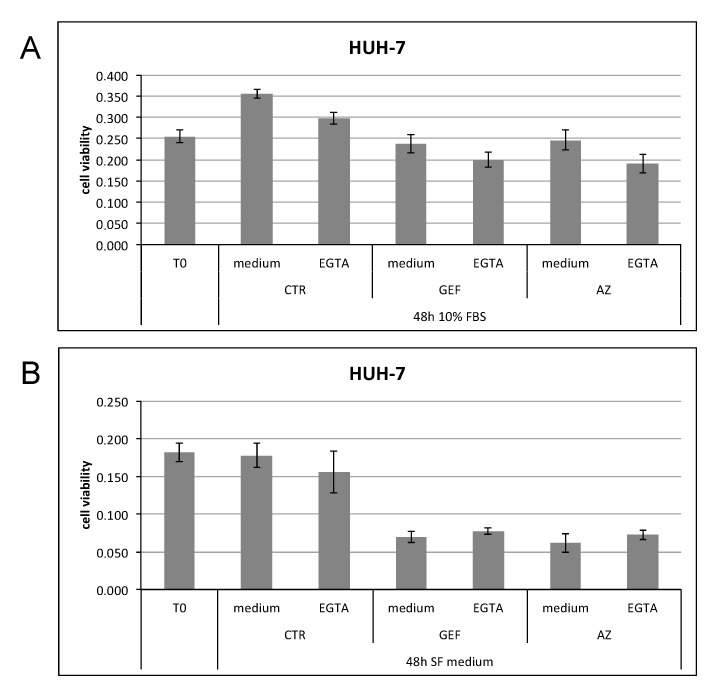
HUH-7 not starved (10% FBS (**A**)) or starved (SF medium (**B**)) cells were left untreated (medium as CTR) or treated with 2 mM EGTA for 48 h in the presence or absence of GEF or AZ.

**Figure 9 cancers-11-01588-f009:**
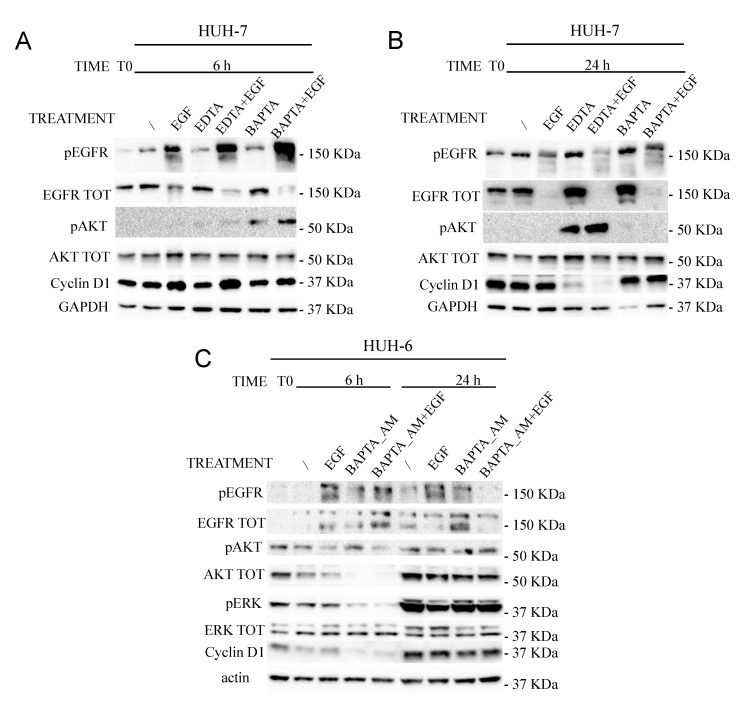
Starved HUH-7 (**A**,**B**) and HUH-6 cells (**C**) (T0) were left untreated (as CTR) or treated with 2 mM EDTA or 10 μM BAPTA_AM with or without 100 ng/mL EGF. The cell signaling cascade was analyzed by western blot after 6 h and 24 h.

**Figure 10 cancers-11-01588-f010:**
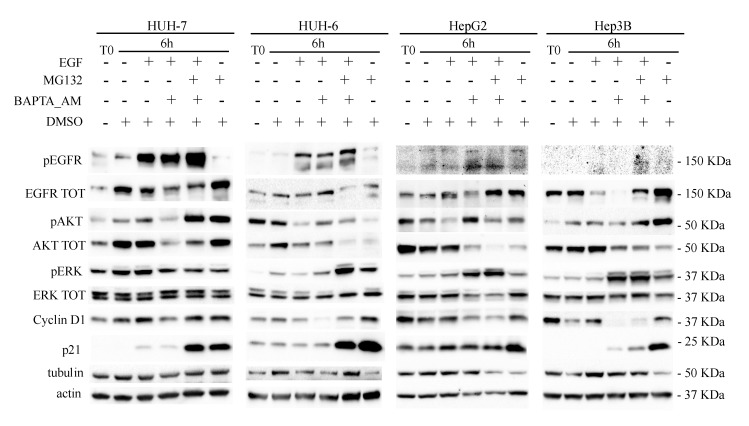
Starved HUH-7, HUH-6, HepG2, and Hep3B cells (T0) were left untreated (as CTR) or treated with 10 μM BAPTA_AM. After 30 min, 40 μM MG132 were added for a further 30 min. 100 ng/mL EGF were added for a total time of 6 h before cells harvesting.

**Figure 11 cancers-11-01588-f011:**
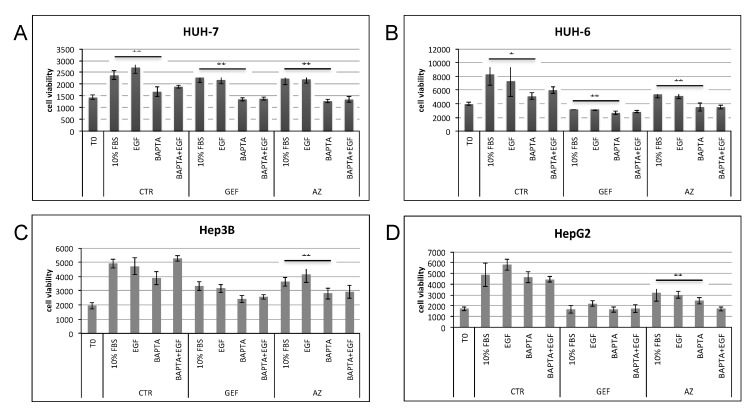
48 h proliferation assay of (**A**) HUH-7, (**B**) HUH-6, (**C**) Hep3B and (**D**) HepG2 cell lines. After 2 h starvation (in 0% FBS medium) cells were treated or not in 10% FBS medium with BAPTA_AM for 1 h. After 1 h, 0.5 × GEF, AZ or DMSO (as CTR) was added to the medium for 48 h. *p* value < 0.05 (*); *p* value < 0.01 (**).

**Table 1 cancers-11-01588-t001:** GEF and AZ IC50 in HCC cell lines after three days incubation.

Cell Line	GEF (µM)	AZ (µM)
HepG2	>25	4.7 ± 0.2
HUH-7	17.7 ± 1.6	5.0 ± 0.1
HUH-6	23.9 ± 0.3	4.5 ± 0.1
Hep3B	22.6 ± 0.07	7.9 ± 0.2

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
