# Peer review of "Calcium Regulates HCC Proliferation as well as EGFR Recycling/Degradation and Could Be a New Therapeutic Target in HCC"

_cancers, 2019, doi:10.3390/cancers11101588_

Round 1
Reviewer 1 Report
The revision addressed previous concerns.
Reviewer 2 Report
Although changes have been made, these appear to be relatively minor and don't constitute a major improvement that would be required for publication in Cancers.
Reviewer 3 Report
The authors have satisfactorily met the reviewers' comments.
This manuscript is a resubmission of an earlier submission. The following is a list of the peer review reports and author responses from that submission.
Round 1
Reviewer 1 Report
In this work, Modica et al. analyze the role of calcium in human hepatoma (HCC) cell lines harboring a wild type form of EGFR, and in particular its role in proliferation and in EGFR downmodulation. The results highlight that calcium is involved in the proliferative capability of HCC cells, as its subtraction is responsible for EGFR degradation by proteasome machinery and, as a consequence, for EGFR intracellular signaling downregulation. They also found that calcium-regulated EGFR signaling is cell line-dependent: in cells responding weakly to EGF, calcium seems to have an opposite effect on EGFR internalization/degradation mechanisms. These results suggest that besides EGFR, calcium could be a new therapeutic target in HCC. While the research described an interesting phenomenon, the conclusions are not well supported by the data shown. Below are some specific points.
In line 114, it is not clear what are “the above-mentioned cell lines”, because no cell lines mentioned above. It is not clear how sequencing of EGFR can demonstrate that EGFR was expressed in the wild form in all the HCC cell lines tested (data not shown). EGF used in the experiment is at a much higher concentration than usually people use. Multiple figure labeling is either too small to read or mis-labelled. What is the concentration of GEF and AZ in the experiments (Figure 2A)? There is no DMSO control in the proliferation assay (Figure 4). There is no statistic analysis in many bar graphs. In Figure 6 and 9, even EGF by itself has no effect on pAKT and pERK. How can the study examine the effect of EDTA and other Ca2+ chelators.
Reviewer 2 Report
The authors present a study of calcium signaling in regards to EGFR using cell lines analyzed in various conditions by westernblot. The results provided are poorly described in the figures that are too many and of poor quality, which is unacceptable for this level of publication. As example multiple figures can’t be read due to the wrong or too small texts, the legends are not detailed enough to interpret the results, the number of experiments performed and the statistical tests are not clearly indicated). This makes the results difficult to navigate and interpret. In addition, no in vivo model or primary tumor work is presented. The overall quality, complexity and level of proof bring up by the study doesn’t reach the requirement for publication in this journal in the present form.
Reviewer 3 Report
In this work the role of calcium in proliferation and EGFR downmodulation was evaluated. The results showed, according to authors’, that calcium is involved in the proliferative activity of HCC cells, its subtraction is responsible for proteasomal degradation of EGFR and EGFR intracellular signaling downregulation. Calcium-regulated EGFR signaling was found to be cell line-dependent. In cells responding weakly to EGF, calcium seems to have an opposite effect on EGFR internalization/degradation mechanisms. These results suggest that besides EGFR, calcium could be a new therapeutic target in HCC.
General comment: The work is often written in verbose form, emphatic and unclear. For example, the explanation of many acronyms is missing, it is not clear what is meant by "calcium signaling". The introduction is very long and its connection with the aims of the work is not clear, the discussion is partially speculative .
Specific comments:
The figures are of poor quality. The inscriptions are too small, the parameter measured must be indicated in the abscissa. It is not clear to what refers the evaluation of significance between two parameters when one of them (i.e. not treated with EGF) is equal to zero. In Fig.3 the immunoblots of the effects of treatment with GEF or AZ IC50 on the expression of different proteins are shown. However, the quantitative evaluation of these effects and the significance of differences is not shown. The Fig.3C shows the growth of different cell lines in the presence of GEF or AZ IC50. However, the growth of untreated controls is not shown and it is not clear to what differences refer the p values. Section 2.3. It s not clear to what “side effects” the authors refer to. Furthermore, EDTA is able to form complexes not only with calcium ion, but also with magnesium, copper, zinc and numerous other ions. Compared to EDTA, EGTA it has a lower affinity for magnesium, but it cannot be excluded that some other ions, in association to calcium, are chelated. The effect of EDTA was evaluated on the viability of only Huh6 and Huh7 cells. However, no statistical evaluation was made. The analysis of the effect of EDTA on EGFR, AKT, ERK cascades and cyclin D1 (Fig.6) was made in only Huh7 cells and, apparently, in a single experiment. Different cell lines must be evaluated and the statistical significance of the results must be shown. The results in Fig.s 7-10 also refer to apparently single experiments e no statistical analysis was made. Only the statistical evaluation of the 48 h proliferation assay of HUH-7, HUH-6, and Hep3B and HepG2 cell lines was made.